# A prospective population-based multicentre study on the impact of maternal body mass index on adverse pregnancy outcomes: Focus on normal weight

**Fieke van Hoorn**[1☯]*, **Leon de Wit**[1☯], **Lenie van Rossem**[2,3], **Marielle Jambroes**[2], **Floris Groenendaal**[4], **Anneke Kwee**[1], **Marije Lamain - de Ruiter**[1], **Arie Franx**[1,3], **Bas B. van Rijn**[1,3], **Maria P. H. Koster**[3‡], **Mireille N. Bekker**[1‡]

1 Department of Obstetrics and Gynaecology, University Medical Centre Utrecht, Utrecht University, Utrecht, The Netherlands, 2 Department of Public Health, Healthcare Innovation, and Medical Humanities, Julius Centre for Health Sciences and Primary Care, University Medical Centre Utrecht, Utrecht University, Utrecht, The Netherlands, 3 Department of Obstetrics and Gynaecology, Erasmus MC, University Medical Centre Rotterdam, Rotterdam, The Netherlands, 4 Department of Neonatology, University Medical Centre Utrecht, Utrecht University, Utrecht, The Netherlands

☯ These authors contributed equally to this work.
‡ These authors also contributed equally to this work.
* F.vanHoorn@umcutrecht.nl

**Data Availability Statement:** All relevant data are within the manuscript and its Supporting information files.

## Abstract

### Background

Maternal body mass index (BMI) below or above the reference interval (18.5–24.9 kg/m²) is associated with adverse pregnancy outcomes. Whether BMI exerts an effect within the reference interval is unclear. Therefore, we assessed the association between adverse pregnancy outcomes and BMI, in particular within the reference interval, in a general unselected pregnant population.

### Methods

Data was extracted from a prospective population-based multicentre cohort (Risk Estimation for PrEgnancy Complications to provide Tailored care (RESPECT) study) conducted between December 2012 to January 2014. BMI was studied in categories (I: <18.5, II: 18.5–19.9, III: 20.0–22.9, IV: 23.0–24.9, V: 25.0–27.4, VI: 27.5–29.9, VII: >30.0 kg/m²) and as a continuous variable within the reference interval. Adverse pregnancy outcomes were defined as composite endpoints for maternal, neonatal or any pregnancy complication, and for adverse pregnancy outcomes individually. Linear trends were assessed using linear-by-linear association analysis and (adjusted) relative risks by regression analysis.

### Results

The median BMI of the 3671 included women was 23.2 kg/m² (IQR 21.1–26.2). Adverse pregnancy outcomes were reported in 1256 (34.2%). Linear associations were observed between BMI categories and all three composite endpoints, and individually for

**Funding:** The RESPECT study was conducted with the support of the Netherlands Organization for Health Research and Development (ZonMw; https://www.zonmw.nl/en/; project no 209020004). The funder had no role in study design, data collection and analysis, decision to publish, or preparation of the manuscript.

**Competing interests:** The authors have declared that no competing interests exist.

pregnancy-induced hypertension (PIH), preeclampsia, gestational diabetes mellitus (GDM), large-for-gestational-age (LGA) neonates; but not for small-for-gestational-age neonates and preterm birth. Within the reference interval, BMI was associated with the composite maternal endpoint, PIH, GDM and LGA, with adjusted relative risks of 1.15 (95%CI 1.06–1.26), 1.12 (95%CI 1.00–1.26), 1.31 (95%CI 1.11–1.55) and 1.09 (95%CI 1.01–1.17).

## Conclusions

Graded increase in maternal BMI appears to be an indicator of risk for adverse pregnancy outcomes even among women with a BMI within the reference interval. The extent to which BMI directly contributes to the increased risk in this group should be evaluated in order to determine strategies most valuable for promoting safety and long-term health for mothers and their offspring.

## Introduction

Normal weight is defined by the World Health Organization (WHO) as a body mass index (BMI) of 18.5 to 25.0 kg/m$^2$. Maternal BMI below or above this reference interval is associated with various adverse pregnancy outcomes such as gestational diabetes mellitus, hypertensive disorders and large-for-gestational-age neonates [1, 2]. These complications are not only relevant for short-term outcomes, but also pose risks for future pregnancies and predispose to long-term maternal health issues like diabetes mellitus and cardiovascular disease [3, 4]. Maternal BMI in pregnancy may furthermore have long-lasting effects on offspring health and is associated with altered susceptibility to non-communicable diseases later in life [5–11].

The epidemic of overweight and obesity is rapidly increasing as a global health issue across all age categories, including pregnant women. While in some countries obesity levels in women of reproductive age are as high as 19% (United Kingdom) and 32% (United States), there are still countries where the rising problem of high BMI is relatively slow, such as the Netherlands, where most women currently have a BMI within the reference interval (18.5–24.9 kg/m$^2$) and the obesity rate is 10% [12–14].

Due to the observed associations between unfavourable BMI and adverse outcomes in the obstetric population, risk stratification and subsequent treatment protocols, as well as scientific research are primarily aimed at those cases where BMI exceeds the WHO thresholds. Studies often do not apply the finer BMI categories as advised by the WHO for public health purposes [15–18], masking possible effects of BMI within the reference interval. In the general non-pregnant population it has been shown that for conditions such as diabetes mellitus, metabolic syndrome and cardiovascular disease, risks are already increased in individuals with a BMI within the reference interval [19–23]. Such a continuous relationship has also been suggested for BMI and several adverse pregnancy outcomes [15, 17]. However, prospective studies addressing the association between BMI and adverse pregnancy outcomes primarily focused on the BMI within the reference interval are lacking. Exploring the full extent of the impact of BMI is crucial in order to develop and implement adequate strategies to address BMI-related morbidity. A moderately elevated risk in the majority of the population could contribute to more adverse events than a small group of high-risk individuals [24]. Small changes in the large low-risk group, although maybe not directly noticeable in the individual, may ultimately

result in large health benefits for the entire population and significant effects on healthcare costs, also known as the prevention paradox [25].

Therefore, in this study we prospectively evaluate the association of BMI and adverse pregnancy outcomes in an unselected cohort of pregnant women in the Netherlands. Our main goal was to determine this association in pregnant women falling within the reference interval for BMI.

## Materials and methods

### Setting and study population

This study was part of the prospective population-based multicentre study the Risk EStimation for PrEgnancy Complications to provide Tailored care (RESPECT) study [26]. The RESPECT study was primarily designed to validate prognostic models for gestational diabetes mellitus and preeclampsia in a regional cohort [26, 27]. Pregnant women were invited to participate by their obstetric health care provider from December 2012 to January 2014 and were included before 14 weeks of gestation. Women were included without specific exclusion criteria to compose a cohort representative of the general obstetric population; all women who were able to provide informed consent in Dutch could participate. Also, recruitment was performed in all type of obstetric care facilities in the central region of the Netherlands (31 independent midwifery practices, five secondary hospitals, and one tertiary care facility). The number of births in our region comprise approximately 20.000 of the annual 170.000 births nationwide.. Participants received routine obstetric care according to Dutch clinical guidelines. For the present study we included singleton pregnancies ≥24 weeks of gestation without chromosomal anomalies. The study was approved by the medical ethics committee of the University Medical Centre Utrecht (protocol number 12-432/C) and all participants provided written informed consent.

### Exposure

Maternal prepregnancy BMI was calculated as weight in kilograms divided by squared height in meters ($kg/m^2$). Both height and prepregnancy weight were collected by self-administered questionnaires in early pregnancy (S1 and S2 Files). We applied the 7-category World Health Organization classification for BMI (I: <18.5, II: 18.5–19.9, III: 20.0–22.9, IV: 23.0–24.9, V: 25.0–27.4, VI: 27.5–29.9 and VII: ≥30.0 $kg/m^2$) [28, 29]. The additional categories used in the 7-category classification are recommended by the WHO to facilitate international comparisons especially regarding public health research and allowed us to assess the association between adverse pregnancy outcomes and BMI within the reference interval (group II-IV; BMI 18.5–24.9 $kg/m^2$) [29, 30]. BMI was furthermore considered as a continuous variable.

### Co-variates

Data concerning baseline characteristics were collected through a set of standardized questionnaires issued to both participants and their healthcare professionals. Gestational age based on the crown-rump-length measurement at ultrasound examination was recorded by the health care provider in the first trimester, along with random venous glucose (mmol/l), and systolic and diastolic blood pressure (mm Hg). Demographics reported by the subject included smoking (yes/no), ethnicity (white (Western European, other Western)/ non-white (African, Hindustani, Moroccan, Turkish, Middle-Eastern, Asian, other non-Western, mixed), educational level (low/medium/high by the International Standard Classification of Education [31, 32]), method of conception (spontaneous/assisted), family history for diabetes mellitus,

hypertension and cardiovascular disease (positive in case of one or more affected first-degree family members) (S1 and S2 Files).

## Outcomes

Pregnancy outcomes were collected postpartum by the health care provider by filling in Case Report Forms. The composite adverse maternal outcome included pregnancy-induced hypertension, preeclampsia, gestational diabetes mellitus (GDM), thromboembolic event, eclampsia, and maternal death. Pregnancy-induced hypertension was defined as systolic blood pressure ≥140 mmHg and/or diastolic blood pressure ≥90 mmHg measured at two consecutive occasions after 20 weeks of gestation in women who previously had a normal blood pressure and preeclampsia if combined with proteinuria (≥300 mg per 24 hours or a protein-creatinine-ratio over 0.30), according to guidelines effective during the study period [33]. GDM was diagnosed with a 75-g oral glucose tolerance test (OGTT) with either a fasting glucose ≥7.0 mmol/l or 2 hour post load ≥7.8 mmol/l (WHO 1999 criteria). According to Dutch guidelines, selective screening is performed in women with predefined risk factors (previous GDM, previous child with birthweight >95[th] percentile, maternal BMI ≥30.0 kg/m$^2$, polycystic ovary syndrome or a first degree family member with diabetes) between 24–28 weeks of gestation. Excessive fetal growth, clinical signs or polyhydramnios were reasons to perform an OGTT in both women with or without risk factors.

The composite adverse neonatal outcome included preterm birth, small-for-gestational-age (SGA) or large-for-gestational-age (LGA) infant, admission to a Neonatal Intensive Care Unit, 5 minute Apgar Score (AS) <7, arterial umbilical artery blood pH <7.0 and perinatal mortality (≥24 weeks of gestation until 7 days postpartum). Preterm birth was defined as a gestational age under 37 weeks. This was further subdivided into spontaneous and iatrogenic preterm birth (induction of labour due to maternal or fetal condition). LGA and SGA were defined as a birthweight above the 90[th] and below the 10[th] percentile respectively, using the Dutch birthweight reference graphs, adjusted for gestational age, parity and neonatal gender [34].

All women in which at least one of the above stated components of adverse maternal or neonatal outcomes occurred were labelled as positive for the composite adverse pregnancy outcome.

## Statistical analysis

Inclusion criteria were applied to the multiple imputed data set with ten imputations, also used in the primary analyses of the RESPECT cohort [26]. Multiple imputation was performed with an imputation model using all exposures, covariates and outcomes to minimize potential bias, because for some participants information was missing and these data were not missing completely at random (S1 Table) [26]. Imputed values were included when calculating descriptive statistics. Analyses were performed on each of the imputed data sets and results were pooled by applying Rubin's rules without any transformation of the estimates.

The rates of adverse pregnancy outcomes per category of the 7-category BMI classification were presented as the number of cases per BMI category divided by the total number of subjects within that BMI category, expressed as a percentage. A linear-by-linear association analysis was conducted to assess whether there was a linear trend between the rates of adverse pregnancy outcomes across the seven ordinal BMI categories.

The association between adverse pregnancy outcomes and BMI were expressed in (adjusted) relative risks with 95% confidence intervals calculated by Cox regression with robust variance with time set as a constant [35, 36]. BMI category III (20.0–22.9 kg/m$^2$) of the 7-category classification served as the reference category to calculate relative risks per category.

To determine the association between BMI and pregnancy outcomes specifically on BMI within the reference interval, (adjusted) relative risks were calculated with BMI as a continuous variable in a sub-analysis only including subjects with a BMI 18.5–24.9 kg/m$^2$. Adverse pregnancy outcomes with less than 75 cases were not assessed individually, but incorporated in the composite outcomes.

We selected confounders using the Directed Acyclic Graphs (DAGs) method in a consensus meeting (FH, LW, BBR and MNB). We identified smoking, educational level and maternal age as confounders and relative risks were adjusted for these variables. An overview of the considered confounders, mediators and competing exposures are presented in S1 Fig.

Analyses were performed using SPSS version 25.0 (SPSS Inc., Chicago, IL).

## Results

The multiple imputed dataset of the RESPECT cohort comprised of 3738 pregnancies of which 3671 were included for analysis in the present study after exclusion of fetal demise <24 weeks of gestation (n = 27), multiple pregnancies (n = 32) and/or chromosomal anomaly in the neonate (n = 15) (Fig 1).

Characteristics of the study population are presented in Table 1. Overall median prepregnancy BMI was 23.2 (IQR 21.1–26.2) kg/m$^2$. The number of subjects in BMI category I to VII were 100 (2.7%), 377 (10.3%), 1242 (33.8%), 738 (20.1%), 553 (15.1%), 297 (8.1%), 364 (9.9%), respectively. The median BMI of the obese subjects in category VII was 32.7 kg/m$^2$ (IQR 31.2–35.4).

Pregnancy outcomes for the cohort are presented in Table 2. Median gestational age at delivery was 40 weeks and 0 days (IQR 39 weeks and 1 days—40 weeks and 5 days) with a

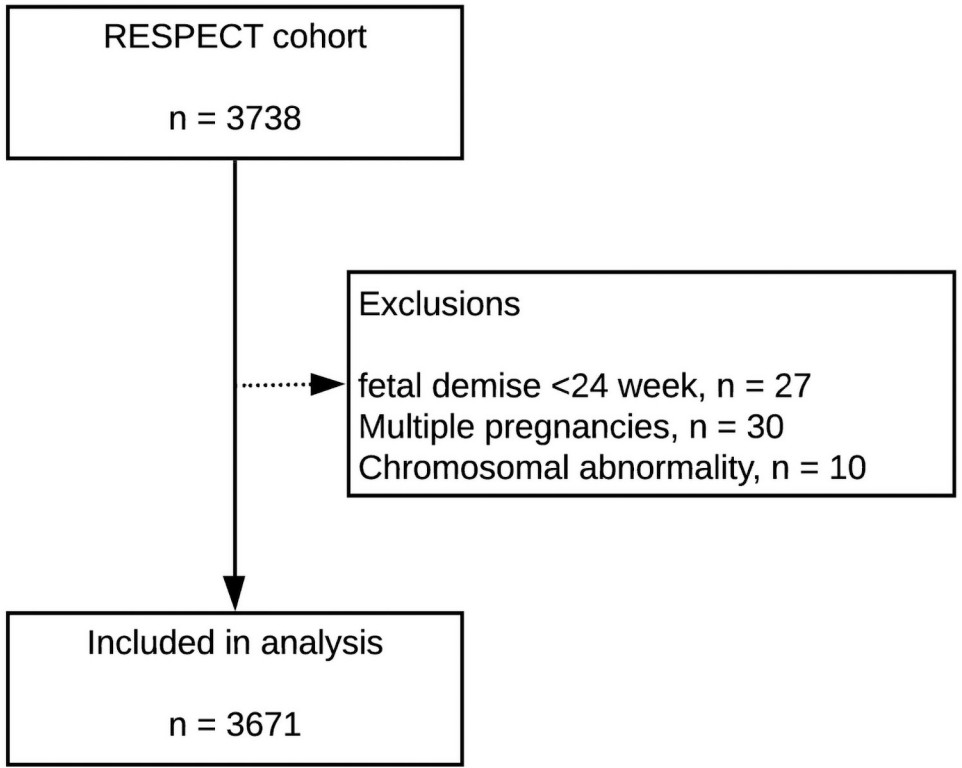

**Fig 1. Flowchart of the study population.**

**Table 1. Characteristics of the study population (n = 3671).**

| | | |
|---|---|---|
| Maternal age (years) [a] | | 30.8 (4.2) [b] |
| Prepregnancy body mass index (kg/m$^2$) | | 23.2 (21.1–26.2) [b] |
| Prepregnancy body mass index category– | I. <18.5 | 100 (2.7) |
| | II. 18.5–19.9 | 377 (10.3) |
| | III. 20.0–22.9 | 1242 (33.8) |
| | IV. 23.0–24.9 | 738 (20.1) |
| | V. 25.0–27.4 | 553 (15.1) |
| | VI. 27.5–29.9 | 297 (8.1) |
| | VII. ≥ 30.0 | 364 (9.9) |
| Blood pressure (mmHg) [a] – | systolic | 115 (12) |
| | diastolic | 67 (8) |
| Glucose (mmol/L) [a] | | 4.7 (4.4–5.1) [b] |
| Ethnicity (white) | | 3339 (91.0) |
| Educational level– | low | 270 (7.4) |
| | medium | 1256 (34.2) |
| | high | 2145 (58.4) |
| Parity (nulliparous) | | 1637 (44.6) |
| Spontaneous conception | | 3417 (93.1) |
| Smoking during pregnancy | | 284 (7.7) |
| Comorbidity [c] | | 126 (3.4) |
| Chronic hypertension | | 60 (1.6) |
| Pre-existent diabetes mellitus | | 13 (0.4) |
| Cardiovascular disease | | 27 (0.7) |
| Thromboembolic event | | 14 (0.4) |
| Other [d] | | 16 (0.4) |
| Positive first degree family history of diabetes mellitus | | 241 (6.6) |
| Positive first degree family history of hypertension | | 357 (9.7) |
| Positive first degree family history of cardiovascular disease | | 265 (7.2) |

Data are number (percentage) unless stated otherwise.

[a] measured at the first prenatal visit.

[b] Data are mean (standard deviation) or median (interquartile range).

[c] Numbers may not add up since women could have more than one comorbidity.

[d] Other comorbidities include kidney disease, antiphospholipid syndrome, and system lupus erythematosus.

mean neonatal birthweight of 3511 ± 578 grams. Pregnancy-induced hypertension was the most frequent adverse maternal outcome (6.6%), followed by GDM (5.0%) and preeclampsia (2.2%). There were no cases of maternal death. Preterm birth occurred in 176 women (4.9%) of which the majority was spontaneous (3.2%). LGA and SGA were present in 13.2% and 7.3% of neonates born in this cohort, respectively. There were 11 cases (0.3%) of perinatal death.

The rates of the composite and individual adverse outcomes showed an increasing proportion of events per incremental BMI category ($p_{for\ trend}$ = <0.05), except for preterm birth and SGA (Fig 2). Furthermore, no linear trend was found when preterm birth was stratified for iatrogenic and spontaneous preterm birth. The observation that the largest absolute number of adverse events occurred in the group with a BMI within the reference interval (category II to IV) was present in all the assessed composite and individual perinatal outcomes.

**Table 2. Pregnancy outcomes in the study population (n = 3671).**

| General pregnancy outcomes | |
|---|---|
| Gestational age at delivery (weeks and days) | 40w0d (39w1d–40w5d) [a] |
| Birthweight (grams) | 3511 (578) [a] |
| Neonatal sex (male) | 1870 (50.9) |
| Adverse pregnancy outcome (composite) [b,c] | 1256 (34.2) |
| Adverse maternal outcome (composite) [c] | 462 (12.6) |
| Pregnancy-induced hypertension | 243 (6.6) |
| Preeclampsia | 79 (2.2) |
| Gestational diabetes mellitus | 183 (5.0) |
| Thromboembolic event | 7 (0.2) |
| Eclampsia | 2 (0.1) |
| Maternal death | 0 (0.0) |
| Adverse neonatal outcome (composite) [c] | 951 (25.9) |
| Preterm birth [d] | 179 (4.9) |
| Spontaneous preterm birth | 117 (3.2) |
| Iatrogenic preterm birth | 62 (1.7) |
| Small-for-gestational-age [e] | 267 (7.3) |
| Large-for-gestational-age [f] | 484 (13.2) |
| Neonatal intensive care unit admission | 64 (1.7) |
| Apgar Score <7 after 5 minutes | 58 (1.6) |
| Arterial umbilical cord blood pH < 7.0 | 38 (1.0) |
| Perinatal death | 11 (0.3) |

Data are number (percentage) unless stated otherwise.

[a] Data are mean (standard deviation) or median (interquartile range).

[b] Consisting of all listed components of adverse maternal outcomes (composite) and/or adverse neonatal outcomes (composite).

[c] Numbers may not add up since women could have more than one adverse pregnancy outcome.

[d] Gestational age <37 weeks.

[e] Birthweight <10th percentile.

[f] Birthweight >90th percentile.

Crude and adjusted relative risks (aRR) for the composite, maternal and neonatal outcomes are shown in Table 3. Compared to reference category III, the aRR for the composite adverse maternal outcome and GDM was significantly increased from category IV onwards, i.e. the highest subgroup within the normal BMI range. The aRR of BMI category IV, V, VI, VII were 1.45 (95%CI 1.06–1.98), 1.99 (95%CI 1.47–2.70), 2.54 (95%CI 1.82–3.55), 3.76 (95%CI 2.83–5.00) for the composite adverse maternal outcome and 2.09 (95%CI 1.22–3.60), 2.52 (95%CI 1.49–4.28), 3.88 (95%CI 2.22–6.79), 6.76 (95%CI 4.19, 10.91) for GDM. A similar graded pattern of increasing aRR per incremental BMI category was found for the composite adverse pregnancy outcome, pregnancy-induced hypertension and LGA, however, with significantly elevated risks only in BMI categories above the reference interval. Although not significant, there was a trend towards higher risk in the highest BMI categories for preeclampsia, whereas the highest risk for SGA was observed in the lowest categories. No pronounced association was found for the composite adverse neonatal outcome and preterm birth.

The association between BMI and pregnancy outcomes in the subpopulation of women with a normal BMI (18.5–24.9 kg/m$^2$) as a continuous variable is shown in Table 4. BMI was significantly positively associated with the risk for the composite adverse maternal outcome,

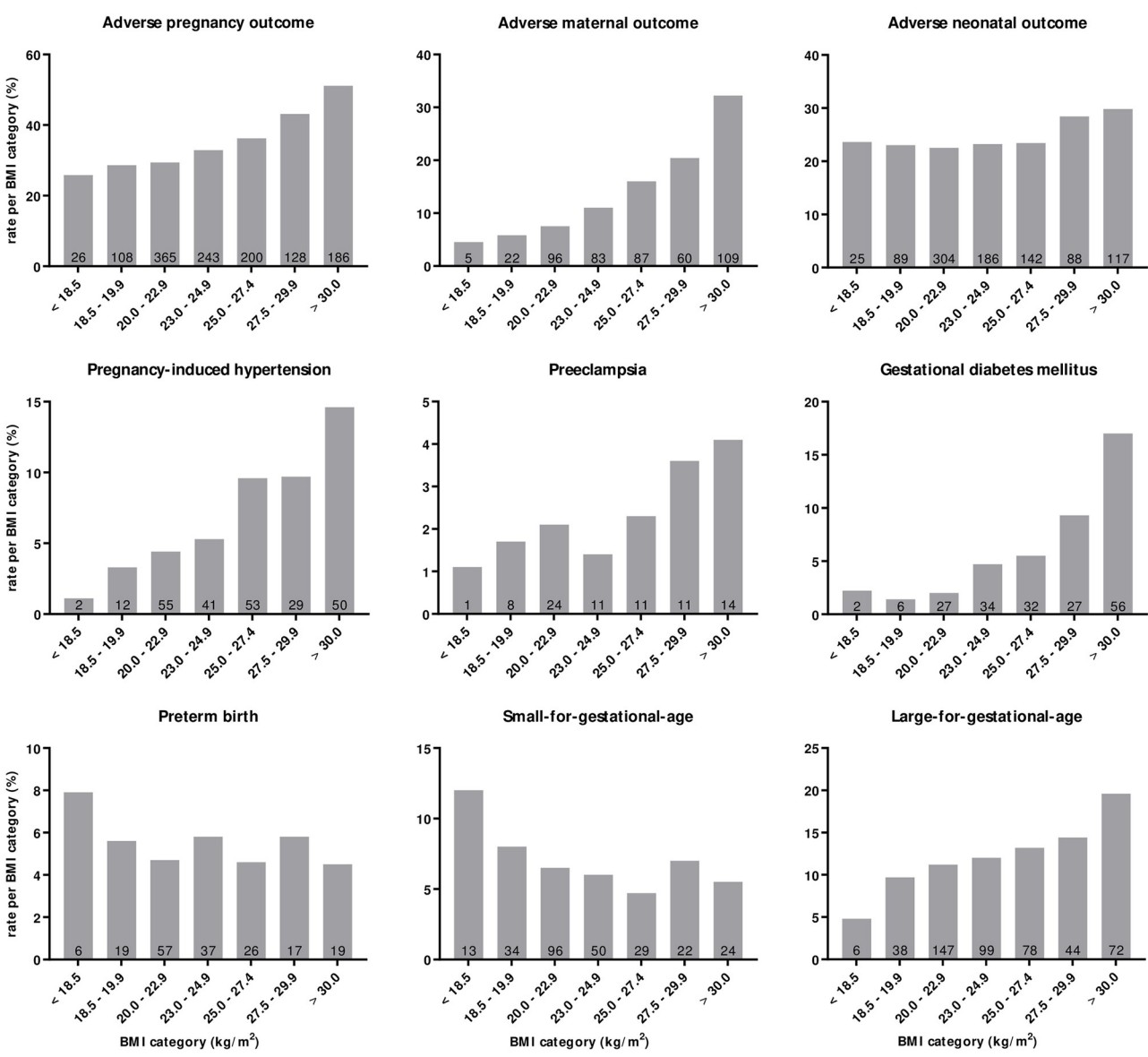

**Fig 2. The rate of pregnancy outcomes and number of events per maternal prepregnancy body mass index (BMI) category.** The height of the grey bars indicate the rate of pregnancy outcomes per BMI category (the number of events per BMI category divided by the total number of subjects within that BMI category; expressed as a percentage). The number in the grey bars represents the number of events per maternal prepregnancy body mass index (BMI) category. Adverse maternal outcome (composite): pregnancy-induced hypertension, preeclampsia, gestational diabetes mellitus, thromboembolic event, eclampsia, maternal death. Adverse neonatal outcome (composite): preterm birth (gestational age <37 weeks), small-for-gestational-age (<10th birthweight percentile), large-for-gestational-age (>90th birthweight percentile), neonatal intensive care admission, Apgar-score <7 after 5 minutes, pH <7.0 in arterial cord blood, congenital anomalies, perinatal death. Adverse pregnancy outcome (composite): adverse maternal and/or neonatal outcome.

pregnancy-induced hypertension, GDM and LGA in this group, with adjusted relative risks of respectively 1.15 (95%CI 1.06–1.26), 1.12 (95%CI 1.00–1.26), 1.31 (95%CI 1.11–1.55) and 1.09 (95%CI 1.01–1.17) per BMI-point increment. Trends were observed for the composite adverse pregnancy outcome and SGA with an aRR of respectively 1.04 (95%CI 0.99–1.09)and 0.91 (95%CI 0.83–1.01) per BMI-point increment. For preeclampsia and preterm birth no association with BMI within the reference interval was found.

**Table 3. Pregnancy outcomes per maternal prepregnancy BMI category (n = 3671).**

**COMPOSITE OUTCOMES**

| BMI category | Adverse pregnancy outcome (n = 1256) | | | | Adverse maternal outcome (n = 462) | | | | Adverse neonatal outcome (n = 951) | | | |
|---|---|---|---|---|---|---|---|---|---|---|---|---|
| | RR | 95%CI | aRR [a] | 95%CI | RR | 95%CI | aRR [a] | 95%CI | RR | 95%CI | aRR [a] | 95%CI |
| I. <18.5 | 0.87 | 0.55–1.40 | 1.40 | 0.53–1.38 | 0.63 | 0.23–1.76 | 0.65 | 0.23–1.82 | 1.03 | 0.64–1.66 | 0.99 | 0.62–1.59 |
| II. 18.5–19.9 | 0.97 | 0.77–1.22 | 1.22 | 0.78–1.23 | 0.77 | 0.48–1.24 | 0.78 | 0.49–1.25 | 0.97 | 0.75–1.25 | 0.97 | 0.75–1.26 |
| III. 20.0–22.9 | | reference | | reference | | reference | | reference | | reference | | reference |
| IV. 23.0–24.9 | 1.12 | 0.94–1.33 | 1.33 | 0.94–1.32 | 1.47 | 1.07–2.01 | 1.45 | 1.06–1.98 | 1.03 | 0.85–1.25 | 1.02 | 0.84–1.24 |
| V. 25.0–27.4 | 1.23 | 1.03–1.47 | 1.47 | 1.02–1.45 | 2.05 | 1.52–2.78 | 1.99 | 1.47–2.70 | 1.05 | 0.85–1.31 | 1.05 | 0.84–1.30 |
| VI. 27.5–29.9 | 1.47 | 1.19–1.80 | 1.80 | 1.17–1.78 | 2.63 | 1.88–3.68 | 2.54 | 1.82–3.55 | 1.22 | 0.95–1.56 | 1.21 | 0.94–1.55 |
| VII. ≥30.0 | 1.74 | 1.45–2.09 | 2.09 | 1.41–2.04 | 3.89 | 2.93–5.16 | 3.76 | 2.83–5.00 | 1.31 | 1.05–1.64 | 1.28 | 1.02–1.61 |

**MATERNAL OUTCOMES [b]**

| BMI category | Pregnancy-induced hypertension (n = 243) | | | | Preeclampsia (n = 79) | | | | Gestational diabetes mellitus (n = 183) | | | |
|---|---|---|---|---|---|---|---|---|---|---|---|---|
| | RR | 95%CI | aRR [a] | 95%CI | RR | 95%CI | aRR [a] | 95%CI | RR | 95%CI | aRR [a] | 95%CI |
| I. <18.5 | 0.35 | 0.05–2.48 | 0.37 | 0.05–2.67 | 0.53 | 0.07–3.92 | 0.50 | 0.07–3.72 | 1.02 | 0.24–4.29 | 1.08 | 0.26–4.55 |
| II. 18.5–19.9 | 0.74 | 0.40–1.38 | 0.74 | 0.39–1.38 | 1.04 | 0.44–2.46 | 1.02 | 0.43–2.42 | 0.69 | 0.26–1.81 | 0.72 | 0.27–1.90 |
| III. 20.0–22.9 | | reference | | reference | | reference | | reference | | reference | | reference |
| IV. 23.0–24.9 | 1.26 | 0.82–1.93 | 1.26 | 0.82–1.94 | 0.76 | 0.34–1.68 | 0.75 | 0.34–1.67 | 2.17 | 1.26–3.73 | 2.09 | 1.22–3.60 |
| V. 25.0–27.4 | 2.14 | 1.46–3.16 | 2.13 | 1.44–3.14 | 0.99 | 0.47–2.12 | 0.96 | 0.45–2.06 | 2.70 | 1.60–4.55 | 2.52 | 1.49–4.28 |
| VI. 27.5–29.9 | 2.22 | 1.40–3.52 | 2.20 | 1.39–3.49 | 1.90 | 0.90–4.03 | 1.82 | 0.86–3.88 | 4.17 | 2.40–7.25 | 3.88 | 2.22–6.79 |
| VII. ≥30.0 | 3.11 | 2.09–4.61 | 3.12 | 2.09–4.67 | 2.06 | 1.05–4.04 | 1.92 | 0.97–3.80 | 7.20 | 4.51–11.51 | 6.76 | 4.19–10.91 |

**NEONATAL OUTCOMES [c]**

| BMI category | Preterm birth (n = 179) | | | | Small-for-gestational-age (n = 267) | | | | Large-for-gestational-age (n = 484) | | | |
|---|---|---|---|---|---|---|---|---|---|---|---|---|
| | RR | 95%CI | aRR [a] | 95%CI | RR | 95%CI | aRR [a] | 95%CI | RR | 95%CI | aRR [a] | 95%CI |
| I. <18.5 | 1.25 | 0.51–3.07 | 1.16 | 0.47–2.86 | 1.66 | 0.88–3.16 | 1.32 | 0.69–2.54 | 0.46 | 0.17–1.23 | 0.49 | 0.18–1.32 |
| II. 18.5–19.9 | 1.08 | 0.59–1.96 | 1.08 | 0.59–1.97 | 1.16 | 0.77–1.75 | 1.18 | 0.78–1.79 | 0.84 | 0.58–1.23 | 0.84 | 0.58–1.23 |
| III. 20.0–22.9 | | reference | | reference | | reference | | reference | | reference | | reference |
| IV. 23.0–24.9 | 1.09 | 0.70–1.72 | 1.08 | 0.68–1.70 | 0.87 | 0.59–1.29 | 0.84 | 0.57–1.24 | 1.14 | 0.86–1.50 | 1.15 | 0.87–1.51 |
| V. 25.0–27.4 | 1.02 | 0.61–1.70 | 0.99 | 0.59–1.66 | 0.68 | 0.43–1.07 | 0.66 | 0.42–1.04 | 1.20 | 0.89–1.60 | 1.21 | 0.90–1.63 |
| VI. 27.5–29.9 | 1.26 | 0.70–2.25 | 1.21 | 0.68–2.16 | 0.98 | 0.61–1.58 | 0.93 | 0.58–1.50 | 1.25 | 0.87–1.80 | 1.27 | 0.88–1.84 |
| VII. ≥30.0 | 1.12 | 0.64–1.98 | 1.05 | 0.60–1.85 | 0.87 | 0.53–1.43 | 0.77 | 0.46–1.28 | 1.67 | 1.24–2.24 | 1.72 | 1.28–2.33 |

[a] aRR was adjusted for age, smoking and educational level.

*BMI*, prepregnancy body mass index. *RR*, relative risk. *95%CI*, 95% confidence interval. *aRR*, adjusted relative risk.

## Discussion

### Principal findings

In this prospective cohort study with a low prevalence of obesity compared with most populations [37, 38], we assessed associations between maternal BMI and several major adverse pregnancy outcomes. Our findings show a graded association between BMI and adverse pregnancy outcomes extending across the BMI range considered 'normal' (18.5–24.9 kg/m$^2$), mostly driven by maternal complications, and to a lesser extent by neonatal complications. In the sub population of women with a BMI within the reference interval, we observed a significant trend of increased relative risk per single BMI point for the composite adverse maternal outcome, pregnancy-induced hypertension, gestational diabetes mellitus, and for large-for-gestational-age. Although relatively small for the individual, on a population level these effects may be substantial, given the large number of women in the normal BMI group.

**Table 4. Associations between maternal prepregnancy BMI as a continuous variable and adverse pregnancy outcomes in a subpopulation of women with a BMI within the reference interval (18.5–24.9 kg/m²).**

| Pregnancy outcome | RR | 95%CI | aRR | 95%CI |
|---|---|---|---|---|
| Adverse pregnancy outcome (composite)[a] | 1.05 | 1.00–1.10 | 1.04 | 0.99–1.09 |
| Adverse maternal outcome (composite) | 1.16 | 1.06–1.26 | 1.15 | 1.06–1.26 |
| Pregnancy-induced hypertension | 1.11 | 0.99–1.25 | 1.12 | 1.00–1.26 |
| Preeclampsia | 0.95 | 0.79–1.13 | 0.95 | 0.80–1.14 |
| Gestational diabetes mellitus | 1.35 | 1.15–1.60 | 1.31 | 1.11–1.55 |
| Adverse neonatal outcome (composite) | 1.023 | 0.97–1.07 | 1.02 | 0.97–1.07 |
| Preterm birth | 0.99 | 0.89–1.11 | 0.99 | 0.88–1.12 |
| Small-for-gestational-age | 0.93 | 0.84–1.02 | 0.91 | 0.83–1.01 |
| Large-for-gestational-age | 1.08 | 1.01–1.17 | 1.09 | 1.01–1.17 |

*BMI*, body mass index; *NA*, not applicable (no cases). *RR*, relative risk. *95%CI*, 95% confidence interval. *aRR*, adjusted relative risk (adjusted for age, smoking and educational level).

[a] Consisting of all components of adverse maternal outcomes (composite) and/or adverse neonatal outcomes (composite).

## Strengths of the study

The novelty of this study is that the examined population effect on maternal and neonatal morbidity focussed on in-group differences within normal-range BMI. Benefits of our study include a population-based design with an unselected population of pregnant women prospectively recruited in both midwifery practices (low risk) and hospital-based antenatal clinics (medium to high risk). While our study population with a majority of women having a BMI within the reference interval (median prepregnancy BMI 23.2 kg/m²), reflecting the current situation in Netherlands, was very suitable for the purpose of our study, we believe our findings are likely to be generalizable to many countries or regions worldwide [14, 30, 39]. The application of the recommended finer WHO classification or BMI makes our research internationally comparable [28, 40].

## Limitations of the data

Limitations include self-reported prepregnancy weight and height. Although a 2018 meta-analysis has shown that self-reported weight and height among women of reproductive age are representative and can be used in clinical and research settings [41], others have found that self-reporting, as well as categorization, can influence the association between weight and adverse pregnancy outcomes [42, 43]. However, data on first-trimester BMI in our cohort, which was registered by the healthcare professional and therefore not dependent on self-reporting, was strongly correlated with prepregnancy BMI ($r$ 0.98; p-value <0.001) and showed a linear relationship through the entire BMI range, including the extremes (S3 File). Also, our analyses on BMI as a continuous variable within the reference interval do not suggest significant misclassification bias. Another limitation is that the study population consists of mostly women of Western European descent with a high educational level. Therefore, our findings need to be interpreted with caution for other populations, and need to be addressed in other more diverse populations. Furthermore, several pregnancy complications that could have been of interest in relation to maternal BMI were not collected in the RESPECT cohort and could therefore not be explored in our study (e.g. post-partum haemorrhage, instrumental vaginal delivery, emergency caesarean section, hyperbilirubinemia, and long-term effects on offspring health).

## Interpretation

In clinical practice, continuous measures are often categorized, as clinicians need thresholds to trigger interventions [24]. The same applies to BMI where previous studies mainly used the four WHO categories (underweight, normal weight, overweight, obesity) and, with that, overlooked possible in-group differences. The impact of BMI $\geq$25.0 kg/m$^2$, i.e. overweight and obesity, on the risk of adverse pregnancy outcomes is unequivocal, as recently summarized in a review of 156 meta-analyses [2]. However, as shown in the 2008 analysis of the HAPO study, many risk factors for adverse pregnancy outcomes, in this case glucose levels, extend well below the clinical thresholds for disease, i.e. gestational diabetes mellitus [44]. In subsequent analyses of the same cohort, similar linear relations were found between maternal BMI and adverse pregnancy outcomes including preeclampsia, neonatal adiposity, fetal hyperinsuline-mia and LGA, independent of maternal glycaemia [15]. In a retrospective study analysing data of the Canadian birth registry, data were also suggestive of a (curvi)linear association between BMI between 17 and 50 kg/m$^2$ and pregnancy complications spanning the full range of BMI, including women classified by the WHO as normal weight [16]. This observation is further supported by data from a Chinese population and a recent individual participant data meta-analysis of 39 cohort studies assessing prepregnancy BMI, and a large British cohort assessing first-trimester BMI [18, 38, 45]. These studies are consistent with our findings of BMI being a strong indicator of risk across the entire population of pregnant women, even in BMI categories not requiring clinical action at an individual level. Our study showed a strong correlation between prepregnancy and first-trimester BMI, and the observed associations of both parameters with adverse outcomes were largely similar (S3 File). This may be explained by the generally limited gestational weight gain in the first trimester.

The mechanisms by which elevated BMI exerts its adverse effects are multifactorial and probably even precede pregnancy itself. Elevated BMI influences metabolic state, including alterations in circulating inflammatory cytokines, hormones and metabolites, which are suggested to negatively affect gamete and embryo quality, early placental function and gene expression [5, 46, 47]. Metabolic changes and altered fetal programming are associated with fetal growth, adiposity and hyperinsulinemia, independent from glycaemic status [15]. Also insulin resistance, dyslipidaemia, chronic inflammation and oxidative stress have all been linked to the development of hypertensive disorders and gestational diabetes mellitus [48, 49]. Additionally, the risk for mechanical complications rises with increasing BMI, resulting in adverse outcomes such as failure to progress, birth injury, increased risk for caesarean section, perioperative difficulties and anaesthesia-related problems [50, 51]. Although it is likely that these mechanisms for BMI-related complications are similar, it is unknown to what extent these mechanisms explain the increased risk in women within the normal BMI range, since the evidence mostly originates from studies in overweight and obese women. Possibly, in addition to BMI, differences in body fat distribution and gestational weight gain may add to the variety observed in women with a normal BMI and their risk for adverse outcomes [52–54].

Population-wide approaches to improve health have been proven beneficial in several lifestyle-dependent conditions in non-pregnant individuals and are increasingly gaining attention. For instance, only modest reduction in mean systolic blood pressure at population level, even in sub populations classified as normotensive, results in lower cardiovascular mortality [55–57]. Also, several countries around the world implemented population-wide initiatives such as taxes on sugar-sweetened drinks and unhealthy energy-dense foods attempting to decrease obesity levels [58–61]. Similar population-wide strategies may also be worth exploring in future studies to reduce BMI-associated perinatal complications. Effects of such interventions might further extent to health improvement in offspring in light of the Developmental

Origins of Health and Disease concept [5], and in the general population by prevention of life-style-dependent conditions. It is important to note that our findings and above proposed health strategies were aimed to respectively explore and counteract the effects of BMI on a population level. Our research should not promote 'fat shaming', nor do we propose strict weight loss regimes for women with a BMI within the reference interval.

In addition, it may also be that the association between normal BMI and negative health outcomes we found in our study is not a direct effect of maternal weight itself, but rather the result of alternative pathways captured by BMI. Previous studies have shown that social determinants of health, such as area of residence, can have profound effects on pregnancy outcomes [62–64]. Such factors, for which BMI may be a potential risk indicator, and their contribution to individual pregnancy health needs further clarification as efforts to reduce their impact would require different approaches than solely weight management.

## Conclusion

Findings from our study indicate that the current thresholds for overweight and obesity may not be sufficient to address the full impact of BMI on adverse pregnancy outcomes. Graded increase in BMI appears to be an indicator of risk for pregnancy complications even among women with a BMI within the reference interval. The extent to which BMI directly contributes to the increased risk in this group, that still comprises the majority of women in our population, should be evaluated in order to determine strategies most valuable for promoting safety and long-term health for future generations of mothers and children.

## Supporting information

**S1 Fig. Directed Acyclic Graph of the relation between body mass index (exposure) and adverse pregnancy outcome (outcome).**
(PDF)

**S1 Table. Baseline characteristics of patients in the RESPECT cohort, stratified by variables that were available for imputation.**
(PDF)

**S1 File. RESPECT-study questionnaire—First visit (translated English version).**
(PDF)

**S2 File. RESPECT-study questionnaire—First visit (original Dutch version).**
(PDF)

**S3 File. Correlation between prepregnancy and first-trimester body mass index.**
(PDF)

**S1 Dataset. Minimal anonymized dataset of the RESPECT cohort.**
(XLSX)

**S1 Checklist.**
(DOCX)

## Acknowledgments

We thank all the pregnant women who participated in the RESPECT study and all members of the RESPECT study group.

## Author Contributions

**Conceptualization:** Fieke van Hoorn, Leon de Wit, Lenie van Rossem, Marielle Jambroes, Floris Groenendaal, Anneke Kwee, Marije Lamain - de Ruiter, Arie Franx, Bas B. van Rijn, Maria P. H. Koster, Mireille N. Bekker.

**Data curation:** Fieke van Hoorn, Marije Lamain - de Ruiter.

**Formal analysis:** Fieke van Hoorn, Leon de Wit, Lenie van Rossem.

**Funding acquisition:** Floris Groenendaal, Anneke Kwee, Marije Lamain - de Ruiter, Arie Franx, Maria P. H. Koster, Mireille N. Bekker.

**Investigation:** Fieke van Hoorn, Leon de Wit, Marije Lamain - de Ruiter, Maria P. H. Koster.

**Methodology:** Fieke van Hoorn, Leon de Wit, Lenie van Rossem, Marielle Jambroes, Floris Groenendaal, Anneke Kwee, Arie Franx, Bas B. van Rijn, Maria P. H. Koster, Mireille N. Bekker.

**Project administration:** Fieke van Hoorn, Leon de Wit, Anneke Kwee, Marije Lamain - de Ruiter.

**Resources:** Anneke Kwee, Marije Lamain - de Ruiter.

**Supervision:** Lenie van Rossem, Arie Franx, Bas B. van Rijn, Maria P. H. Koster, Mireille N. Bekker.

**Validation:** Arie Franx, Maria P. H. Koster, Mireille N. Bekker.

**Visualization:** Fieke van Hoorn, Leon de Wit.

**Writing – original draft:** Fieke van Hoorn, Leon de Wit, Lenie van Rossem, Bas B. van Rijn, Maria P. H. Koster, Mireille N. Bekker.

**Writing – review & editing:** Fieke van Hoorn, Leon de Wit, Lenie van Rossem, Marielle Jambroes, Floris Groenendaal, Anneke Kwee, Marije Lamain - de Ruiter, Arie Franx, Bas B. van Rijn, Maria P. H. Koster, Mireille N. Bekker.

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
