## [Decision Letter · Decision Letter 0]

29 Jun 2021

PONE-D-21-01714

A prospective population-based multicentre study on the impact of maternal body mass index on adverse pregnancy outcomes: focus on normal weigh

PLOS ONE

Dear Dr. van Hoorn,

Thank you for submitting your manuscript to PLOS ONE. After careful consideration, we feel that it has merit but does not fully meet PLOS ONE’s publication criteria as it currently stands. Therefore, we invite you to submit a revised version of the manuscript that addresses the points raised during the review process.

We look forward to receiving your revised manuscript.

Kind regards,

Diane Farrar

Academic Editor

PLOS ONE

Journal Requirements:

1. Please ensure that your manuscript meets PLOS ONE's style requirements, including those for file naming. The PLOS ONE style templates can be found athttps://journals.plos.org/plosone/s/file?id=wjVg/PLOSOne_formatting_sample_main_body.pdf and https://journals.plos.org/plosone/s/file?id=ba62/PLOSOne_formatting_sample_title_authors_affiliations.pdf

3. In your Methods section, please provide additional information about the participant recruitment method and the demographic details of your participants. Please ensure you have provided sufficient details to replicate the analyses such as:

- a description of any inclusion/exclusion criteria that were applied to participant recruitment

- a statement as to whether your sample can be considered representative of a larger population

- a description of how participants were recruited.

6. Please amend either the title on the online submission form (via Edit Submission) or the title in the manuscript so that they are identical.

Additional Editor Comments :

Please expand your description of the multiple imputation carried out including how any potential bias was minimised 

Please remove 'Adjusted relative rates' from the title of table 3 it is unnecessary

Reviewers' comments:

Reviewer's Responses to Questions

**Comments to the Author**

1. Is the manuscript technically sound, and do the data support the conclusions?

Reviewer #1: Yes

Reviewer #2: Partly

Reviewer #3: Yes

2. Has the statistical analysis been performed appropriately and rigorously? 

Reviewer #1: Yes

Reviewer #2: Yes

Reviewer #3: I Don't Know

3. Have the authors made all data underlying the findings in their manuscript fully available?

Reviewer #1: Yes

Reviewer #2: No

Reviewer #3: Yes

4. Is the manuscript presented in an intelligible fashion and written in standard English?

Reviewer #1: Yes

Reviewer #2: Yes

Reviewer #3: Yes

5. Review Comments to the Author

Reviewer #1: This is a very well written and very clear manuscript covering an important matter.

My only concern is about language and more specifically about the use of the adjective “normal” throughout the manuscript and in the title. I would suggest to check with the “language matters guideline” as it seems that classing something as normal can be frowned upon. Maybe this needs to be discussed with the editor. I would suggest to change to describe ranges of BMI [18.5-24.9] rather than “normal BMI”.

In the discussion “cut-off point for normal versus abnormal”, I would just state that clinicians need thresholds to trigger interventions.

One thing missing from the discussion is how authors chose studied adverse outcomes. The adverse studied outcomes were limited to hypertension, preeclampsia, GDM , thromboembolic event eclampsia and maternal death. I imagine that other adverse outcomes such as post-partum hemorrhage were not studied as they were not available in their dataset. This needs clarification.

Reviewer #2: I would like to congratulate the authors for this needed paper that has been presented in an intelligent fashion with all needed statistical analysis.

However, I have a few queries:

1)Please include how the education levels (low to high) were classified.

2)In the legend of table 1, a,b,c were described but b is nowhere to be found in the table.

3)Line 187-188, authors state: "composite and individual adverse outcomes showed an increasing proportion of events per incremental BMI category (p for trend = <0.05), except for preterm birth and SGA (Fig 2)". However in the figure, several of the other graphs (GDM, adverse neonatal outcome, preeclampsia) did not follow the described trend.

4)There seem to be a major error in many of the graphs in figure 2. The y axis shows percentage but seems wrong. For eg: for adverse neonatal outcome:- for the 1st bar 25events/951events=2.6%; 2nd bar 89/951=9.3%, 3rd bar 304/951=32% but for around 24%, 23% and 22%, respectively, are shown in the figure.

Reviewer #3: INTRODUCTION: Few grammatical errors. Page 8,lines 136-137, the definition of preterm birth is not clear. Line 138,what do the authors mean by induction of pregnancy?

RESULTS: Table 1- what do the authors mean by maternal age and parity at intake? Table 2, The abbreviation NICU should be explained somewhere in the text.

6. PLOS authors have the option to publish the peer review history of their article (what does this mean?). If published, this will include your full peer review and any attached files.

Reviewer #1: **Yes: **Caroline Diguisto

Reviewer #2: No

Reviewer #3: No

---

## [Author Response · Author response to Decision Letter 0]

28 Jul 2021

F. van Hoorn, M.D.

University Medical Centre Utrecht

Utrecht University

Lundlaan 6

3584 EA, Utrecht

The Netherlands

28 July 2021

Dear Diane Farrar, 

Thank you for considering our manuscript “A prospective population-based multicenter study on the impact of maternal body mass index on adverse pregnancy outcomes: focus on normal weight.” [PONE-D-21-01714] for publication in PLOS ONE. Please find below our responses to the points raised by the academic editor and reviewers. Along with our responses, we have submitted a marked-up, as well as a unmarked version of our revised manuscript. 

Thank you in advance for your re-considering our manuscript. 

Should you require any further information, please do not hesitate to contact me. 

On behalf of all authors, 

Sincerely, 

Fieke van Hoorn, M.D. 

Department of Gynaecology and Obstetrics 

University Medical Centre Utrecht, University of Utrecht 

The Netherlands 

F.vanHoorn@umcutrecht.nl

 

Journal requirements

We have checked and revised the manuscript concerning style requirements, including those for file naming.

We have added the questionnaire that was filled out by pregnant women after their first prenatal visit as supporting information. We added referrals to this questionnaire as appropriate in the manuscript. The original questionnaire was issued to participants in the Dutch language (S2 File); the English version serves as supporting information to this publication only (S1 File). 

3. In your Methods section, please provide additional information about the participant recruitment method and the demographic details of your participants. Please ensure you have provided sufficient details to replicate the analyses such as:

- a description of any inclusion/exclusion criteria that were applied to participant recruitment

- a statement as to whether your sample can be considered representative of a larger population

- a description of how participants were recruited.

We added the additional information as requested to the ‘setting and study population’ paragraph of the methods.

“Pregnant women were invited to participate by their obstetric health care provider from December 2012 to January 2014 and were included before 14 weeks of gestation. Women were included without specific exclusion criteria to compose a cohort representative of the general obstetric population; all women who were able to provide informed consent in Dutch could participate. Also, recruitment was performed in all type of obstetric care facilities in the central region of the Netherlands (31 independent midwifery practices, five secondary hospitals, and one tertiary care facility). The number of births in our region comprise approximately 20.000 of the annual 170.000 births nationwide.”

We have added a supplemental file (S3 File) in which the correlation between prepregnancy BMI and first-trimester BMI is shown. Data on first-trimester BMI is also found in the shared database.

We have uploaded a minimal anonymized dataset as requested (S1 dataset).

6. Please amend either the title on the online submission form (via Edit Submission) or the title in the manuscript so that they are identical.

We amended the title in the online submission form to “A prospective population-based multicentre study on the impact of maternal body mass index on adverse pregnancy outcomes: focus on normal weight”.

We have reviewed our reference list, and found it to be complete, correct and not to include any retracted papers that we are aware of.

Additional Editor Comments:

1. Please expand your description of the multiple imputation carried out including how any potential bias was minimised.

We expanded our description of the multiple imputation below. The supplemental table (S1 Table) was already included in the initial submission.

“Inclusion criteria were applied to the multiple imputed data set with ten imputations, also used in the primary analyses of the RESPECT cohort [26]. Multiple imputation was performed with an imputation model using all exposures, covariates and outcomes to minimize potential bias, because for some participants information was missing and these data were not missing completely at random (S1 Table) [26].”

2. Please remove 'Adjusted relative rates' from the title of table 3 it is unnecessary

We removed this phrase form the title of table 3 as requested.

Review Comments to the Author: 

Reviewer #1

This is a very well written and very clear manuscript covering an important matter.

1. My only concern is about language and more specifically about the use of the adjective “normal” throughout the manuscript and in the title. I would suggest to check with the “language matters guideline” as it seems that classing something as normal can be frowned upon. Maybe this needs to be discussed with the editor. I would suggest to change to describe ranges of BMI [18.5-24.9] rather than “normal BMI”.

We originally used the terms “normal weight”, “normal BMI” and “BMI within the normal range” because these are stated in the globally used BMI classification by the World Health Organisation. We value the reviewers view and feel that using alternative termination is perfectly in line with the message of our manuscript. We found a relevant paper by Whyte et al 2018 on this topic in which the term “normal range” is discussed in the context of laboratory results and the preferred term “reference interval” is proposed (doi: 10.1136/postgradmedj-2018-135983). Therefore we altered “normal weight”, “normal BMI” and “BMI within the normal range” to “BMI within the reference interval” throughout the entire manuscript. We would suggest not to alter the title, because of length and clarity, but we did change the short title into “BMI within the reference interval and pregnancy outcome”.

2. In the discussion “cut-off point for normal versus abnormal”, I would just state that clinicians need thresholds to trigger interventions.

We altered this sentence into: “In clinical practice, continuous measures are often categorized, as clinicians need thresholds to trigger interventions [24].”.

3. One thing missing from the discussion is how authors chose studied adverse outcomes. The adverse studied outcomes were limited to hypertension, preeclampsia, GDM , thromboembolic event eclampsia and maternal death. I imagine that other adverse outcomes such as post-partum hemorrhage were not studied as they were not available in their dataset. This needs clarification.

Indeed several important and relevant outcome measures have not been collected in the prospective RESPECT cohort used for this study. Although the outcomes that were reported include the most common and/or the most severe outcomes, outcomes such as post-partum haemorrhage, instrumental vaginal delivery and emergency caesarean section rates would also have been of interest. We have integrated this statement in the ‘limitations of the data’ section of the discussion within the sentence where other neonatal complications of interest were discussed:

“Furthermore, several pregnancy complications that could have been of interest in relation to maternal BMI were not collected in the RESPECT cohort and could therefore not be explored in our study (e.g. post-partum haemorrhage, instrumental vaginal delivery, emergency caesarean section, hyperbilirubinemia, and long-term effects on offspring health).”

Reviewer #2: 

I would like to congratulate the authors for this needed paper that has been presented in an intelligent fashion with all needed statistical analysis. However, I have a few queries:

1. Please include how the education levels (low to high) were classified.

Pregnant women reported the highest level of education they had finished in the first trimester questionnaire (added as supporting information). The International Standard Classification of Education (ISCED) was applied, as stated in reference no. 32: low = less than primary, primary and lower secondary education (levels 0-2); medium = upper secondary and post-secondary non-tertiary education (levels 3 and 4); high = Tertiary education (levels 5-8). We added that we used the International Standard Classification of Education in the “covariates” section of the methods.

2. In the legend of table 1, a,b,c were described but b is nowhere to be found in the table.

We have added ‘b’ after ‘comorbidity’ in table 1. 

Please note that after another revision in this table ‘b’ has now become ‘c’.

3. Line 187-188, authors state: "composite and individual adverse outcomes showed an increasing proportion of events per incremental BMI category (p for trend = <0.05), except for preterm birth and SGA (Fig 2)". However in the figure, several of the other graphs (GDM, adverse neonatal outcome, preeclampsia) did not follow the described trend.

The p for trend test examines whether there is a linear increase or decrease between ordinal categories. We agree with the reviewer that the bar graphs for some outcomes with a significant p for trend do not display a perfect linear shape. As we look at the bar graphs, this is mostly caused by the extreme categories. These categories comprise the smallest numbers of participants and weighed less is this crude measurement. In a population including more pregnant women with an extreme BMI it could be further explored whether the association between these outcomes and BMI are indeed linear or whether they follow a different trend (e.g. exponential, J- or U-shaped).

4. There seem to be a major error in many of the graphs in figure 2. The y axis shows percentage but seems wrong. For eg: for adverse neonatal outcome:- for the 1st bar 25events/951events=2.6%; 2nd bar 89/951=9.3%, 3rd bar 304/951=32% but for around 24%, 23% and 22%, respectively, are shown in the figure.

In figure 2 the outcome rate (event rate, absolute risk) per BMI category is depicted as the number of events within that category divided between the number of women within that same category. So for adverse neonatal outcome, the first bar (<18.5kg/m2): 25 events in a total of 100 women, resulting in an event rate of 25% in this category. This is explained in the methods (line 151-153): “The rates of adverse pregnancy outcomes per category of the 7-category BMI classification were presented as the number of cases per BMI category divided by the total number of subjects within that BMI category, expressed as a percentage.” To clarify this matter we added this information to the legend of figure 2.

Reviewer #3: 

1. INTRODUCTION: Few grammatical errors. Page 8,lines 136-137, the definition of preterm birth is not clear. Line 138,what do the authors mean by induction of pregnancy?

Preterm birth was defined as a gestational age under 37 weeks. This was explained in the next sentence after line 137. “Induction of pregnancy” should have stated “induction of labour” and has been adjusted in the manuscript.

2. RESULTS: Table 1- what do the authors mean by maternal age and parity at intake? Table 2, The abbreviation NICU should be explained somewhere in the text.

By ‘at intake’ is meant ‘measured at the first prenatal visit’ which is added as footnote a to table 1 for clarification. The abbreviation NICU was replaced by ‘neonatal intensive care unit’. 

General:

While revising your submission, please upload your figure files to the Preflight Analysis and Conversion Engine (PACE) digital diagnostic tool, https://pacev2.apexcovantage.com/. PACE helps ensure that figures meet PLOS requirements.

We have uploaded our images to PACE digital diagnostic tool with no problems encountered.

---

## [Editor Report · Decision Letter 1]

9 Sep 2021

A prospective population-based multicentre study on the impact of maternal body mass index on adverse pregnancy outcomes: focus on normal weight

PONE-D-21-01714R1

Dear Dr. van Hoorn,

We’re pleased to inform you that your manuscript has been judged scientifically suitable for publication and will be formally accepted for publication once it meets all outstanding technical requirements.

Kind regards,

Diane Farrar

Academic Editor

PLOS ONE

---

## [Editor Report · Acceptance letter]

13 Sep 2021

PONE-D-21-01714R1 

A prospective population-based multicentre study on the impact of maternal body mass index on adverse pregnancy outcomes: focus on normal weight 

Dear Dr. van Hoorn:

I'm pleased to inform you that your manuscript has been deemed suitable for publication in PLOS ONE. Congratulations! Your manuscript is now with our production department. 

Kind regards, 

on behalf of

Dr. Diane Farrar 

Academic Editor

PLOS ONE